# Potential of the Red Alga *Dixoniella grisea* for the Production of Additives for Lubricants

**DOI:** 10.3390/plants10091836

**Published:** 2021-09-04

**Authors:** Antonio Gavalás-Olea, Antje Siol, Yvonne Sakka, Jan Köser, Nina Nentwig, Thomas Hauser, Juliane Filser, Jorg Thöming, Imke Lang

**Affiliations:** 1Algae Biotechnology, Institute of EcoMaterials, Bremerhaven University of Applied Sciences, An der Karlstadt 8, D-27568 Bremerhaven, Germany; agavalasolea@hs-bremerhaven.de (A.G.-O.); thauser@mein.gmx (T.H.); 2Center for Environmental Research and Sustainable Technology (UFT), Department Chemical Process Engineering (CVT), University of Bremen, Leobener Straße 6, D-28359 Bremen, Germany; asiol@uni-bremen.de (A.S.); koeser@uni-bremen.de (J.K.); thoeming@uni-bremen.de (J.T.); 3Center for Environmental Research and sustainable Technology (UFT), Department General and Theoretical Ecology (ÖKO), University of Bremen, Leobener Straße 6, D-28359 Bremen, Germany; sakka@uni-bremen.de (Y.S.); nentwig@uni-bremen.de (N.N.); filser@uni-bremen.de (J.F.)

**Keywords:** extracellular polymeric substances, *Dixoniella grisea*, bio-additive, polysaccharides, proteins, fatty acids, red algae, culture conditions, ecotoxicological effect

## Abstract

There is an increasing interest in algae-based raw materials for medical, cosmetic or nutraceutical applications. Additionally, the high diversity of physicochemical properties of the different algal metabolites proposes these substances from microalgae as possible additives in the chemical industry. Among the wide range of natural products from red microalgae, research has mainly focused on extracellular polymers for additive use, while this study also considers the cellular components. The aim of the present study is to analytically characterize the extra- and intracellular molecular composition from the red microalga *Dixoniella grisea* and to evaluate its potential for being used in the tribological industry. *D. grisea* samples, fractionated into extracellular polymers (EPS), cells and medium, were examined for their molecular composition. This alga produces a highly viscous polymer, mainly composed of polysaccharides and proteins, being secreted into the culture medium. The EPS and biomass significantly differed in their molecular composition, indicating that they might be used for different bio-additive products. We also show that polysaccharides and proteins were the major chemical compounds in EPS, whereas the content of lipids depended on the separation protocol and the resulting product. Still, they did not represent a major group and were thus classified as a potential valuable side-product. Lyophilized algal fractions obtained from *D. grisea* were found to be not toxic when EPS were not included. Upon implementation of EPS as a commercial product, further assessment on the environmental toxicity to enchytraeids and other soil organisms is required. Our results provide a possible direction for developing a process to gain an environmentally friendly bio-additive for application in the tribological industry based on a biorefinery approach.

## 1. Introduction

As blue biotechnology has become an emerging field globally, marine resources such as algae are now being targeted for biotechnological applications [1,2]. The need to identify algae-derived molecules is evident given their bioactive potential as, for example, proven anti-inflammatory, antioxidant or antimicrobial effects [3]. With an increasing global market of 7.4% per year, microalgae products reached a market of 1 billion euros during the period of 2016–2018. Moreover, the projected growth of up to 80% by 2024 is sufficient reason to continue the exploration of microalgae as a sustainable and bio-based source for a rich plethora of effective molecules [4].

Besides the already established production of microalgae for food, feed and cosmetic industries, microalgae display alternatives to replace fossil fuel-derived chemicals, for example as surfactants, emulsifiers or lubricants [5,6,7].

Lubricants are a mixture of an oily or watery base liquid and additives. The most important property of lubricants is viscosity, which determines the thickness of the lubricating film and thus the performance of the lubricant [8]. Moreover, viscosity changes as a function of temperature, pressure and shear rate. The lubricant’s additives can be active in the lubricant itself, i.e., improving dispersion and viscosity and functioning as an antioxidant. They can also be surface active as anticorrosive, anti-wear or extreme pressure additives [8]. Current bio-based target molecules as lubricants are polymers such as polysaccharides (PS) and proteoglycans, which are regarded as biodegradable and generally as non-toxic. At present, the food and cosmetic industry widely use polymers as thickeners, stabilizers and hydrogels [9,10,11,12,13]. More recently, biogenic polymers have also received attention as potential lubricants in oil recovery and drilling processes as well as tribological applications [14,15,16,17,18,19].

The success of lubricity seemed to depend directly on the adsorption of the polymers to metal surfaces and was interlinked with the molecular weight, the molecule’s structure and its functional groups. Particularly, sulfur groups are identified to mediate anti-wear protection and enhance lubricity [8,20].

Many microalgae and cyanobacteria, in particular those living in the benthic zone, excrete large amounts of polymeric mucilage which cover their cells. In liquid culture, a minor fraction of the excreted polymers dissolve into the surrounding medium, whereas the majority of polymers remain attached to the cell. Presumably, these polymers render osmo-protection and protect cells from predators [21,22] and viral infections [23].

A biotechnologically important source of algae-based PS are species of cyanobacteria, diatoms and green and red microalgae. Exopolysaccharides of the cyanobacterium *Cyanothece epiphytica* showed excellent potential as a biolubricant [14]. This was related to the similarity of the measured visco-elastic properties of the EPS to conventional grease, showing a high storage modulus compared to the loss modulus (G’ >> G”). These properties are supposed to stabilize the lubricant film thickness when high pressures occur, e.g., in rolling bearings of a high load [14]. High viscosities are considered favorable for lubrication as viscosity controls the lubrication film thickness [24]. Arad et al. [25] state that at high pressures, high loads and low sliding velocities, the main friction mechanism is boundary lubrication. They also stressed that the adhesion of red microalgae polysaccharides, that was related to also present glycoproteins, was an important advantageous influence on the lubrication compared to the properties of hyaluronic acid alone. Interestingly, strong lubricating boundary layers were reported by Lin et al. [26] when using hyaluronic acid together with phosphatidyl choline lipids for tendons. This strong effect was related to the also-present glycoprotein lubricin. Borah et al. [14] also corroborated the versatility of exopolysaccharides, showing their great potential as emulsifiers, flocculants and dispersers. Gasljevic and coauthors [15] evaluated the polysaccharides of several marine microalgae as suitable drag-reducing additives for naval applications. They found that the red microalgae species *Porphyridium cruentum* and *Rhodella maculata*, and the green microalgae species *Schizochlamydella capsulata* and *Chlorella stigmatophora* exhibited the best drag-reducing ability among the strains tested. They also included cellular polysaccharides into their study, which revealed similar properties than the extracellular polymers (EPS), and when applied together, increased drag-reducing ability by fourfold [15]. The potential of PS from red microalgae in tribological processes was superior to the conventional hyaluronic acid as a lubricant in terms of friction reduction, adsorption and stability [25,27]. Notably, only low polymer concentrations were necessary to result in high viscosity [28].

The EPS of red microalgae comprise mainly sulfated PS, and several species from fresh and brackish water as well as seawater habitats have been studied in detail [29]. The unique properties of sulfated PS initiated many research activities to find out more about their chemical composition, physicochemical properties and biosynthesis [28,30,31,32,33]. In general, sulfated PS are negatively charged, and the prominent monosaccharides were found to be xylose, glucose and galactose, which fall into different ratios, depending on growth conditions [20,34,35,36]. In addition to other sugars in smaller proportions and the sulfate content (1–9% *w*/*w*) already mentioned, proteins covalently and non-covalently bound to the PS represent an important component [37,38,39,40]. The protein moiety in microalgal EPS can either have a structural function or act as extracellular enzymes that are involved in the degradation of polymers [21]. However, functional data of proteins within microalgal extracellular polymers are scarce.

With regard to tribological processes, proteins function well as bio-additives. In the food industry, they are not only applied as nutritional additives, but in particular as techno-functional additives, where they fulfil roles as solvents, emulsifiers and thickeners, among other things [41]. Amino acid side chains, particularly sulfur groups, result in differences in various chemical properties such as polarity or reactivity and improve, mixed into a lubricant, the interaction with the surface of the friction partners in the contact zone [42]. Current funded research initiatives are evaluating crop-plant-derived proteins as bio-additive in lubricants. However, even though it is a renewable resource, the sustainability of using lubricant additives derived from crops is questionable in the light of circular economies. Therefore, microorganisms, and especially microalgae are favored. Recently, a study on the use of amphiphilic fungal hydrophobins as aqueous lubricant additives showed effective reduction in friction forces [43]. However, the use of microalgae proteins as lubricant additives represents a widely unexplored field with high potential.

Beside molecules that render lubricity, dispersion and viscosity index improvement, other synthesized and often environmentally toxic substances are used as additives in tribological processes [8]. To name a few of them, biocides, anti-corrosives, antioxidants and extreme pressure additives are commonly supplemented to improve the lubricant performance and extend equipment lifetime. Structural similarities of these conventional additives are found in fatty acids (FA), pigments and amino acid derivatives, among other things. These compounds are synthesized de novo by microalgae, which advances them a suitable resource for the exploitation of bio-additives to replace mineral-oil based additives.

With regard to the use of microalgae fatty acids (FAs) as lubricant additives, only little is known. However, the use of algae oil as lubricant has been described in a recently filed patent [6]. Thus, it can be assumed that FAs synthesized by algae are functionally similar to the FAs already used from vegetable oil [44]. Both water-soluble and lipophilic pigments from microalgae can act as antioxidant and anti-corrosive agents, and are widely applied in different industries, especially in the food, cosmetic and health sectors [45,46]. Plant-based carotenoids and chlorophyll have successfully been tested as octane-boosting additives in gasoline, demonstrating the possibility of using plant-based pigments as additives in chemical industry processes [47]. Nonetheless, little information is available on the use of microalgal pigments as lubricant additives.

Irrespective of the substance group, the replacement of conventional lubricant additives by algal-based ones is also related to the possibility to replace toxic or non-degradable substances by non-toxic degradable ones. In general, red algae are not reported to produce toxins [48], and *D. grisea* EPS was reported to promote cell growth, and possess antimicrobial, antiviral and antioxidant activity [49,50,51,52]. Even though this represents potentially interesting properties for additives, it also indicates a high reactivity, which in turn could induce unwanted side effects when released in the environment.

The purpose of this study was to determine whether the red microalgal strain *D. grisea* UTEX 2320 may be suitable as a source for bio-additives applied in tribological processes. With this interest, we studied the following fractions in terms of chemical composition and ecotoxicity: cells, culture medium containing all exudated material with and without algal cells, and precipitated extracellular polymers. To accomplish this, two different procedures were carried out. First, cultures were separated into cells and medium by centrifugation. This approach was used to account for various culture conditions and harvesting times and tested for their effect on *D. grisea* total content of proteins, sugars and lipids by colorimetric methods. Second, precipitation of EPS was induced by the addition of 2-propanol following Khattar et al. [53] using a ratio of 1:1 (*v*:*v*) (see details in Section 4). These extracellular polymers, the remaining medium with cells and a centrifuged cell pellet were analyzed in more detail for their composition of sugars, proteins and lipids—based on the hypothesis that harvesting time would affect the composition of the respective fractions. In addition, the viscosity of the polymers was investigated and an ecotoxicological screening of the fractions analyzed was performed.

As previously mentioned, data on the molecular composition of sulfated PS in *D. grisea* and also its production under different conditions are available [20,35,37]. However, we aimed for a holistic approach by evaluating the complete culture of *D. grisea*, including its environmental impact, and therefore addressing the need of a biorefinery approach in algae biotechnology. Furthermore, we expected to obtain a comprehensive overview of cellular and extracellular molecules that could function as additives. It was further expected that the results on the ecotoxicological effect of the different culture fractions guide us towards an environmentally friendly production process.

## 2. Results

### 2.1. Cultivation of D. grisea UTEX 2320

The aim of industrial use of *D. grisea* demanded the optimization of the production of both cellular and exudated material. As a consequence, cultivation times were elongated to enable the observation of maximum growth periods or declines in the culture. However, the cultures strongly increased in viscosity with increasing time, resulting in various problems: first, aeration of larger culture vessels had to be adapted to maintain carbon and oxygen levels, as well as mixing of all parts of the culture. Second, centrifugation to separate cells and medium was possible for younger cultures and after precipitation of EPS only (see Section 2.4 for details). As a result, the EPS analyzed after centrifugation and after precipitation differ in the investigated material: EPS obtained after centrifugation covers the whole exudated material that is dissolved freely in the medium (Figure 1). The EPS obtained after the addition of 2-propanol includes all molecules that were able to precipitate. The same is also true for the cell fraction: centrifuged cells may still be covered by an EPS shell or parts of that, while cells centrifuged after addition of 2-propanol are likely to lack this shell. Prior to the precipitation step, centrifugation was not successful for these cells, while it worked well afterwards.

Consequently, we defined the following fractions to include these potential differences in the chemical analysis: medium-C and cells-C are samples separated by centrifugation. EPS-P, medium-P and cells-P were obtained after the addition of 2-propanol for precipitation.

### 2.2. Effect of Culture Conditions and Harvesting Time

Aside from other factors, the salt content of the medium strongly affects algal growth. However, the genus *Dixoniella* occurs in limnic as well as in marine environments [54,55,56,57]. Therefore, a first experiment aiming at defining the optimal salt content for *D. grisea* UTEX 2320 growth was run. The cultivation experiment revealed a clear impact of salt concentration on the cell growth of *D. grisea* (Figure 2). The growth rate at a low salt concentration of 8.25 practical salinity units (psu) was almost twice as high (0.025 OD day^−1^) as the growth rate measured in the culture growing in fresh water medium BG-11 (0.015 OD day^−1^). Higher concentrations of salt, 16.5 psu and 24.75 psu, showed a negative effect on growth (Figure 2). Thus, BG-11 medium with a salinity of 8.25 psu was selected as the optimal culture media for growing *D. grisea* for the next experiments.

To continue the optimization of growth and production of released molecules, we performed an experiment to examine the impact of cultivation temperature on the productivity of *D. grisea*. For this, the cultures were grown in illuminated, temperature-controlled bench-top photobioreactors with a culture volume of 0.4 L. We evaluated the growth at two different temperatures, 20 °C and 25 °C, and we analyzed the EPS content of this culture.

After ten days of cultivation, *D. grisea* reached an OD_750nm_ of 0.7 at 25 °C (0.98 g L^−1^), whereas at a lower temperature of 20 °C, the OD_750nm_ was 0.6 (0.84 g L^−1^, Figure 3a). The extracellular polysaccharide content was higher at 25 °C (0.85 ± 0.08 g_sugars_ g_dry biomass_^−1^, Figure 3b) than at 20 °C (0.60 ± 0.02 g_sugars_ g_dry biomass_^−1^), while proteins released to the medium (0.6 ± 0.16 g_proteins_ g_dry biomass_^−1^) were lower than at 20 °C (0.65 ± 0.05 g_proteins_ g_dry biomass_^−1^).

A final experiment with the addition of an external carbon source was executed to evaluate the performance of this red microalgae (Figure 4). In addition, we determined total sugars, lipids and proteins in both fractions, cells and medium (cells-C and medium-C). The addition of sodium bicarbonate implied a reduced growth after 14 days of cultivation, while the culture without the carbon source continued to grow until the end of the experiment (28 days). The OD_750nm_ in the final day was 2.9 (9.11 ± 0.11 g dry biomass L^−1^) and 3.9 (10.91 ± 0.05 g dry biomass L^−1^) for the photobioreactors with and without sodium bicarbonate addition, respectively. The culture growing without bicarbonate exhibited higher overall production of polysaccharides with a percentage of 56% of ash free dry weight (AFDW), resulting in a highly viscous culture after 28 of cultivation. Of this, 39% represented cellular and 17% released polysaccharides. The same culture also showed a higher total protein content with 10% cellular proteins and 7% released proteins of AFDW. The addition of sodium bicarbonate resulted in slightly lower total polysaccharides (52% of AFDW), but a higher percentage of released polysaccharides (21% of AFDW) compared to the culture with bicarbonate addition. Total protein content was similar with around 8% cellular proteins of AFDW. In both experiments, lipid content was similar with 4–5% AFDW (Figure 4c–d). 

### 2.3. Viscosity

When precipitated EPS (EPS-P) was solved in water, it formed a rather viscous solution. In contrast, cell samples solved without any visible effect on viscosity. Due to the targeted use as bio-additive, we performed additional viscosity measurements with this EPS-P.

The viscosities of the 0.5% EPS solutions for both cultures showed non-Newtonian shear thinning behavior at shear rates between 1 and 1000 s^−1^ (Figure 5). There was no indication for a Newtonian plateau at low shear rates.

### 2.4. Composition of Algal Fractions

To be able to identify the most useful part of the culture for tribological applications, supernatant with EPS (EPS-P), medium-P (mixture of alcohol extractant and culture media), and cells-P were separated and analyzed individually for their molecular composition in detail. In total, nine batches were produced and analyzed (Table 1).

The quantification of sugars, proteins and lipids, as well as an analysis of their composition was performed by GC-MSD and HPLC-DAD. Prior to this, we compared frequently used methods in pilot experiments and chose the most suitable ones based on their recovery (Table 2).

The amino acid content of the focused EPS-P, medium-P and cells-P was analyzed via HPLC-DAD using a pre-treatment according to the methodology described by Frank and Powers [58] with 0.5 M perchloric acid and precolumn online derivatization with *o*-phthalaldehyde (OPA) and fluorenylmethoxycarbonyl (FMOC) as described by Palaniswamy et al. [59] The sugar and protein contents were determined using GC-MSD after silylation with trimethylsilyl ether (TMSE) as described previously [59,60,61,62].

For each substance group, internal standard substances were chosen and used to assess the corresponding recovery (a) during the derivatization process, and (b) during the complete sample preparation process labelled “total work up” (Table 2). Recoveries for derivatization were always higher than 70% of the nominal concentration of the corresponding substance while recoveries of total work up clearly indicate an effect of the sample preparation process on the distribution of the substance groups into the different kinds of samples (EPS-P, cells-P, medium-P). However, this distribution was in accordance with their chemical nature and the amount added initially to the sample, so that the recovery met the expectations.

Results of quantification of monosaccharides, amino acids, and fatty acids extracted from the red alga *D. grisea* are presented in Table 3, Table 4 and Table 5. All results of the algal contents are presented as a percentage of the dry weight of the analyzed sample. In general, the concentration of the analyzed compound groups in the three culture fractions changed with culture conditions and age of the culture batches.

During analysis by GC-MSD, sugars occurred as mono- and disaccharides, as their corresponding trimethylsilyl ether (TMSE) derivatives, as oxime, as alcohol or in dehydrated or alkylated form (Table 3, Table 4 and Table 5). As the sample preparation required for this analysis could have been responsible for the occurrence of these variances of monosaccharides, all forms were considered equally as representatives for the corresponding monosaccharide in Table 3, Table 4 and Table 5 (for details, see Appendix A).

The precipitated EPS contained in total: sugars, in all described forms, from 4 to 272 mg g^−1^ (7.8–37.5%), proteins from 65 to 461 mg g^−1^ (10.5–23.3%) and lipids from 4 to 180 mg g^−1^ (13.5–29.5%; including the glycerol content from 4 to 185 mg g^−1^ (8–19%), Table 3 and Table 4).

In EPS-P, galactose and glucose as oximes were the main sugar components in the range from 50 to 70%. Minor findings of monosaccharides included arabinose (Ara), xylose (Xyl), mannose (Man), ribose (Rib) (about 0.1–4.5%), anhydrosaccharides (0.1–2%), sugar acids (about 0.1–3.4%), methylated sugars (0.5–1.5%), disaccharides (0.1–1.5%) and sugar alcohols (0.1–6.5%). The total amount of amino acids (AA) in the EPS-P of *D. grisea* ranged from 65 to 461 mg g^−1^ (10.5–23.3%). The main components were: lysine (Lys), phenylalanine (Phe), iso-leucine (Ile), tyrosine (Tyr), valine (Val), cysteine (Cys), proline (Pro) and hydroxyproline (OH-Prol) as well. Additional findings of AS traces were: aspartic acid (Asp), glycine (Gly), alanine (Ala), arginine (Arg), leucine (Leu), histidine (His) and tryptophan (Trp).

Analyzed cellular residues (cells-P) contained in total: sugars, in all described forms, from 50 to 783 mg g^−1^ (7.8–32.4%), proteins from 185 to 561 mg g^−1^ (14.5–31.1%) and lipids from 0.7 to 266 mg g^−1^ (13.5–28.5%, including the glycerol content from 20 to 530 mg g^−1^ (10.6–22.5%), Table 5 and Table 6). The differences observed between the different analyses are within the range of variation of the different batches.

Similar to the EPS fraction, galactose and glucose as oximes were the main sugar components in the cell fraction and ranged from 20 to 65%. Minor findings of free sugars in the cells included arabinose (Ara), xylose (Xyl), ribose (Rib), mannose (Man), fucose (Fuc), gulose (Gul) (about 0.5–1.7%), sugar acids (GlcUA about 1.7–2.6%), methylated sugars (0.1–1.2%), and sugar alcohols (2.5–9.0%). Additionally, anhydrosaccharides such as levoglucosan and galactose and glucose disaccharides were detected (3.5–12.4%). The total amount of amino acids in the cells of *D. grisea* ranged from 185.6 to 562 mg g^−1^ (14–31%, see Table 6). The main components were: lysine (Lys), phenylalanine (Phe), iso-leucine (Ile), tyrosine (Tyr), aspartic acid (Asp), valine (Val), cysteine (Cys), proline (Pro) and hydroxyproline (OH-Pro). Additional findings of amino acids traces were: glycine, alanine, arginine, glutamine, glutamic acid, leucine, methionine and tryptophane. Aside from these mayor groups, we also identified traces (0.2–0.5%) of saturated, unsaturated and branched hydrocarbons, terpenes (phytanes) and alcohols with chain lengths from C12 to C16.

In EPS-P and cells-P, the identified compound groups were similar, but differed in relative mass distribution. EPS consisted mainly of sugar compounds and proteins, but was of low lipid content, cells were mainly composed of sugars and lipids and lower in proteins. Due to these findings, the remaining media (medium-P) of the red alga *D. grisea* were additionally analyzed to clarify the contents of the water fraction.

A dialysis step was conducted for these media samples to remove remaining salts. Still, in our case, it also removed amino acids and sugars from the sample (Table 7) and thus increased the relative amount of fatty acids. Consequently, these results were not included in the overall analysis of molecular composition of media samples.

The not-dialyzed media samples (medium-P) from the batches 5, 6 and 7 contained in total: sugars, in all described forms, from 230 to 588 mg g^−1^ (34–54%), proteins from 369 to 461 mg g^−1^ (42–57%) and lipids from 49 to 60 mg g^−1^ (4.5–8.8%, including the glycerol content from 79 to 119 mg g^−1^ (7.2 to 17%) (see Table 7).

For the medium-P, galactose and glucose as oximes were again the main sugar components in the range of 27 to 41%. Findings of free sugars included mannose, ribose and arabinose (about 3.2–115%). Minor ingredients were sugar acids (about 0.1–3.4%), methylated sugars (0.1–0.5%) and sugar alcohols (0.3–4.4%). The total amount of amino acids in the media of *D. grisea* ranged from 10–25%. The main components were the same as described above. We also identified again traces (2.3–11.5%) of saturated, unsaturated and branched hydrocarbons, terpenes (isoprenes and phytanes) and alcohols with chain lengths from C12 to C16. Dialysis of the supernatant medium overnight to reduce the salt content resulted in a clear loss of the soluble compounds except of the lipid fraction.

### 2.5. Ecotoxicological Effects of Algal Samples

Ecotoxicological tests ensuring environmental friendliness needed to be fast and be able to deal with small sample sizes. Furthermore, they should represent different environments and worst-case scenarios, i.e., a high exposure scenario. For this purpose, all test organisms were exposed to the algae sample via aquatic test media. In total, four standard test organisms were chosen: the waterflea *Daphnia magna*, the springtail *Folsomia candida*, the potworm *Enchytraeus crypticus* and the soil and sediment bacterium *Arthrobacter globiformis*.

All organisms were able to survive in the chosen media during the test, so control survival was used as reference. In total, lyophilized fractions obtained after precipitation of seven different batches were tested with all four test organisms. The effects of the tested lyophilized fraction differed with the test organism used: bacterial enzyme activity was not negatively affected by any of the tested fractions or batches, and except for the EPS-P of batch 3, the same was true for daphnids (Table 8). The effect to enchytraeids and collembola differed mainly with the sample type: for enchytraeids, EPS-P samples were highly toxic and caused 100% immobilization, except for the extraction of batch 1. In contrast, the lyophilized medium-P fraction gained from the same batch caused moderate toxicity to enchytraeids (31 ± 13% of test animals; Table 8) while all other lyophilized cells-P and medium-P samples, i.e., fractions containing algal cells, were not toxic to enchytraeids. The collembolan *F. candida* showed the most diverse response when exposed to lyophilized algal fractions. Overall, algal cell-containing samples were toxic when solved in DMSO, but not when solved in medium. However, the negative effect of DMSO solved algal samples to collembola ranged from control toxicity levels (<10% of test animals) to moderate toxicity levels of 25 ± 38% of immobilized animals (Table 8). Out of the five EPS-P batches tested, only two were toxic to *F. candida* (Table 8).

## 3. Discussion

### 3.1. Challenges for Large-Scale Production of D. grisea

To address the need for low salt load for the analytical studies (preventing any damage to GC-MS and other instruments), we aimed at reducing the salinity of the culture medium. The genus *Dixoniella* appears euryhaline, as species distribution is described from freshwater to seawater habitats [54,55,56,57]. In the case of strain UTEX 2320, the best growth was obtained with a salinity of 8.25 psu and no growth was observed at salinities higher than 16.5 psu, approving its brackish water origin. Additionally, freshwater cultivation was not possible with that particular strain.

As previously described, *D. grisea* UTEX 2320 was found to be a microalga with a high potency to produce very viscous cultures [20,37,54]. This kind of behavior is common in high molecular weight polymers and physical gels and other structured interconnected systems [20]. The viscosities were a little lower but in the same order of magnitude, as reported by Liberman and coworkers [20]. This deviation may be a result of our more direct 2-propanol extraction method to produce the tested EPS-P.

The growth behavior of this red microalga challenges the culturing and handling, as well as the down-streaming process for the extraction of valuable compounds. To obtain purified polysaccharides, different membrane filtration techniques have been proposed, but this separation process poses some challenges, such as biofouling and the loss of valuable components in the permeate [63]. Additionally, the centrifugation procedure used in other studies [20] was not always successful in our case. Consequently, alternative separation methods will also be required. The separation into three fractions after direct extraction with 2-propanol may also be used for production on a larger scale. Still, it represents an additional working step and as discussed below, the separation procedure may affect the composition of the obtained fraction itself.

Even though overall biomass accumulation in our air-sparged cultures was comparably low, the degree of viscosity affected the monitoring of biomass dry weight and cell numbers. Furthermore, the aeration flow rate and bubble size impacted the handling of the cultures: *D. grisea* UTEX 2320 formed superficial compact foam with increased flow rates (>250 mL min^−1^) in our system. Thus, for initial characterization of the molecular composition, we focused on a stable and constant biomass and EPS production rather than on high biomass productivity, keeping in mind that the molecular ratio might change with improved growth conditions as previously described [35,64].

Enhanced productivity of *D. grisea* was already achieved by using a sleeve-type reactor system [65], but the biomass obtained from these cultivations was not used for the presented data here. Nevertheless, the production enhancement of the overall biomass productivity of *Dixoniella* is mandatory in order to achieve our goals for the production of multiple products as bio-additives for tribological processes.

Considering the overall challenges for large-scale microalgae production, such as production costs and capacity, a high overall productivity is required to be competitive with other bioproducts [66]. Current photobioreactor systems in use might not be suitable for large scale production of highly viscous cultures, as the mass transfer capacity is significantly lowered. Alternatively, a continuous harvest or a biofilm reactor might be more applicable. This would also provide solutions to the still very high costs of microalgae harvesting and product extraction [32,67,68,69].

### 3.2. Molecular Composition of Dixoniella

To evaluate whether *D. grisea* is a suitable microalga to gain bio-additives for tribological processes, the molecular culture composition was examined. This was done in several independent cultivations and by varying culture parameters, such as temperature, time of harvest (culture age) and addition of a supplementary carbon source (sodium bicarbonate). The impossibility to centrifuge large volumes of *D. grisea* due to its viscosity, resulted in two different sampling procedures that were analyzed for their molecular composition: small volumes (up to 1 mL of sample) were centrifuged (resulting in cells-C and medium-C), and big volumes were extracted by 2-propanol and split into three different fractions EPS-P, cells-P and medium-P.

Looking into the detailed analysis of the different compound groups that was used for the larger volumes (EPS-P, cells-P, medium-P), the initial experiments were dedicated to finding a suitable analytical method that enables the identification of a broad variety of molecules. The sample preparation using the corresponding FAMEs [70,71,72,73] was successful for all kind of our alga fractions (data not shown). The faster silylation to produce trimethylsilyl (TMS)-esters gave stable results and was well suited for sugar analysis. As sugars have been frequently considered the most important fraction in EPS [74,75,76], we used this method for the detailed analysis of all algal culture fractions. GC-MS after derivatization has also been used by other researchers to identify the monosaccharides in EPS [20]. In our case, it also worked well for pellet and media samples.

In general, the data obtained for medium-C revealed a high content of released polysaccharides and proteins that were secreted to the medium in all cultures that were analyzed. The same was observed for medium-P in the GC-MS analysis, while EPS-P samples showed a higher proportion of lipids. These differences suggest that the separation process may have affected the distribution of the different substance groups into the different fractions, in this case, medium-C vs. medium- and EPS-P. However, it is also possible that the glycerol also included in the lipid content came from different sources: it can result from bacterial degradation of sugars, as well as from lipid metabolism [77]. To differentiate these two sources, additional measurements and experiments would be required. Still, an effect of the separation process on EPS composition would allow modifications of the gained product in accordance with the needs of the targeted application. In this case, it would represent an additional advantage for the commercial use of *D. grisea*.

The secretion of released polysaccharides (and other metabolites) and its accumulation in the surrounding medium was concomitant with increasing cell density in the growth experiments. This is a commonly observed phenomenon with polymer-secreting microalgae and cyanobacteria, and is well-studied [33,75]. Especially in this late growth phase, carbon flux is directed to the EPS rather than to other cellular processes. We found extracellular proteins associated with released polysaccharides with a relative percentage of 22–30% of the total medium-C secreted. The ratio changes of released polysaccharides and proteins in the extracellular matrix can be explained by differences in cultivation conditions. As previously described by Ivanova and coworkers, released polysaccharide content seems to be strongly correlated with an increase in temperature [78]. In another study, Soanen and coworkers showed that temperature affected the production of released polysaccharides in the red microalga *P. marinum* [79]. They demonstrated that already a small rise in temperature by 4 °C increased the production of released polysaccharides by almost 2-fold, and with culture media modifications up to 4-fold.

Analyzing the cellular *D. grisea* fraction, we found high amounts of PS and proteins, whereas lipids were less prominent in cells-C samples. When cells are centrifuged after the precipitation step (cells-P), they contain about one third of sugar, proteins, and lipids, each. Taking also the faster separation process of centrifugation and the shorter time period until harvesting for younger, so less viscous, cultures into account, these two major fractions, PS and proteins, should be the focus for developing a biorefinery process, whereas the lipophilic fraction might be a valuable side product.

Even though the relative proportion of sugars varied with the culture component analyzed, the dominant monosaccharides remained galactose and glucose for EPS-P, cells-P and media-P. The highest diversity of monosaccharides was observed in cellular samples while media samples were composed of glucose and galactose and their related oximes only. Aside from monosaccharides, glucose and galactose and their oximes, traces of arabinose, xylose and mannose were also detected. Additionally, methylated sugars, sugar alcohols and sugar acids were found to a smaller extend.

In addition, after static dialysis, most of the water-soluble components could not be detected in the remaining sample, while this was possible for extracted EPS in other studies [80]. As the dialysis was run with the media fraction after EPS extraction by precipitation with 2-propanol, it is possible that the larger polysaccharides were included in the precipitate and the remaining ones were too small to be retained. According to Patel et al. [80] ultrafiltration/diafiltration through a 10 kDa NMWCO membrane at constant pressure of two bars increased purity of extracted EPS in terms of total sugar content, so it is possible that smaller saccharides or attached proteins were removed by dialysis in their experiments as well.

Unlike Libermann and coworkers [20], who identified xylose and rhamnose as major components and did not report either glucose or galactose to be present in their samples, we did not find these sugars in large proportions in *D. grisea* EPS. As the same strain (UTEX 2320) was used in both studies, this difference is either caused by a difference in culture media and conditions or in the EPS extraction method. As a brackish water culture medium was used in both studies, aeration mode or temperature regime remain influencing factors being responsible for the difference in EPS composition. In our case, variations of light intensity and carbon supply (batch 5 and 6, respectively) did not alter EPS composition. As reported elsewhere [81], alterations of temperature, irradiance, pH, nitrogen or phosphate content of the media could change the amount of EPS produced by the red algae *Rhodella violacea*, but had no effect on EPS composition. Thus, it is not very likely that culture medium or culture conditions were the cause for the different EPS composition in *D. grisea* observed in the two studies. Thus, it is more likely that the method used for EPS extraction caused the difference in sugar composition observed here. A similar effect has been observed for the extraction of fucoidan [13]. Libermann and coworkers [20] used a protocol to separate EPS that focuses on a physical separation of EPS and cells by centrifugation [82]. Centrifugation of cultures to separate cells from EPS was not possible in the present study due to the high viscosity of the harvested medium. The use of 2-propanol and cooling to induce precipitation of the extracellular polymers is thus very likely to have altered the measured polymer composition. This is further supported by the observed differences in molecular composition of medium-C and EPS- medium-P discussed above. For the monosaccharides in fucoidan, an effect of the extraction procedure on the detected monosaccharide composition was reported as well [13]. A more precise description of the extraction procedure is thus important for future comparisons in EPS composition.

As mentioned previously, the protein content in EPS-P was slightly lower compared to the cells and clearly lower compared to the medium-P. Interestingly, its composition was also different from the cells, indicating that proteins present in the EPS-P expectedly differ from those in the cells (cells-P).

In summary, the analytical results indicate that (a) EPS and biomass differ significantly in their molecular composition and may thus be used for different purposes, (b) PS and proteins are the main components in all fractions and might represent the main raw materials for bio-additives and (c) further work on the extraction process is needed to develop a simple, but more uniform method that is able to extract target substances while removing the salt from the culture media from the product.

### 3.3. Ecotoxicological Impact of D. grisea Fractions

The replacement of conventional and often toxic lubricant additives by algae-based additives requires the ecotoxicological reviews according to the REACh recommendations. Therefore, we conducted standardized tests with all algae fractions using the waterflea *D. magna*, the potworm *E. crypticus*, the springtail *F. candida*, and the soil bacterium *A. globiformis*, respectively.

Overall, the two soil organisms, *F. candida* and *E. crypticus* were more sensitive towards the lyophilized algal fractions compared to the aquatic test organism and the bacteria. In addition, enchytraeids were most affected by EPS, while Collembola were more frequently affected by algal fractions containing algae cells or parts of them.

The toxicity to *F. candida* varied strongly between the different batches, but was mostly zero when test medium was used as solvent. This suggests a lipophilic substance of the protocol being responsible, as these would only be present in the case of DMSO-solved samples. However, we were not able to identify a single lipophilic substance present in most culture samples that was not found in EPS samples. Still, two fatty acids were found most frequently: palmitic and stearic acid. Unsaturated fatty acids are especially known to act toxic to various aquatic organisms [83]. Additionally, saturated fatty acids can be toxic, even though most studies show a lower toxicity compared to their unsaturated counterparts [84,85]. Ikawa et al. [83] suggest interruption of cell membrane functioning and/or metabolic malfunction as potential reasons for fatty acid toxicity. It is possible that palmitic and stearic acid were able to interrupt the integrity of *F. candida* cuticles and thus caused immobilization when *F. candida* were exposed to cellular-based samples solved in DMSO. On the other hand, DSMO itself could be responsible for the toxicity by enabling hydrophilic substances to enter the cells of *F. candida.* For silver nitrate, a detergent mixture of Tween 20 and TAGAT^®^TO enhanced the silver toxicity to *F. candida* [86]. In this case, a hydrophilic substance which is present in cellular based samples acts toxic to *F. candida* once it enters its tissues. A direct toxicity of DMSO as also described for the detergent mixture [86] can be excluded in the present study, as only tests with valid solvent control, i.e., no additional mortality, were considered for this article. Still, also two EPS samples were toxic to *F. candida*. Therefore, it remains possible that the same substance or substance group is responsible in all cases of *F. candida* toxicity.

Toxic effects to other organisms have only been reported for EPS, not for algal cell samples: for example, microbial pathogens were affected by *D. grisea* EPS, but not by cellular components [50]. However, not all pathogens were affected by *D. grisea* and the effect differed with species. This is also similar to the differences observed in this study: EPS is most toxic to *E. crypticus*, partly toxic to *F. candida*, and not toxic to *D. magna* or *A. globiformis*. Additionally, *Porphyridium* sp. polysaccharide showed moderate cell growth inhibition at a concentration of 1 g L^−1^ [51]. In addition, the red macroalgae *Champia parvula* was able to inhibit the growth of larvae of the dengue mosquito vector *Aedes aegypti* [87]. However, none of these studies linked the observed effects to specific EPS components or some related effect such as altered pH, salinity changes, osmolarity problems or similar. We could rule out effects by remaining bacteria, salinity or pH by additional controls or control experiments (data not shown). Consequently, we expect one component of the EPS or its combination to be responsible for the negative effects observed here.

Overall, lyophilized algal fractions obtained from *D. grisea* were not toxic, if EPS were not included and the sample did not contain DMSO. When EPS is used as an additive for lubricants, further investigation on the toxicity to enchytraeids and other soil organisms is required. This information is especially important, as REACh recommends tests with soil organisms only for high amounts and substances of high lipophilicity [88]. This study does not only contribute to the ecotoxicological evaluation of microalgae-based products, it also illustrates the importance of soil organisms for this evaluation.

### 3.4. Evaluation of Dixoniella for a Biorefinery Approach

Prior work has documented the success of using of bio-based additives for tribological processes, mainly in food and cosmetic industry. Proteins, for example, are techno-functional components in many food products by enhancing dispersion and functioning as emulsifier [21,41]. Additionally, for polysaccharides, in particular those with functional groups, and lipids, the potential as lubricant and additive is evident [14,15,25,44,63]. However, the majority of these studies have either been focused on other organisms, such as bacteria or macroalgae, or have only focused on secreted polymers. In the study by Gasljevic and co-authors, the advantage of combined use of cellular and extracellular polysaccharides to increase the effectiveness as additive has already been shown [15].

In this study, we aimed at expanding the potential product range of *D. grisea* UTEX 2320 by following the biorefinery concept. With this concept, the initial algae biomass is fractionated into multiple intermediates, such as proteins, sugars and lipids, and then further converted to the targeted products [19]. The need for such approaches is obvious in the light of a circular economy and economical challenges faced by algae biotechnology.

The richness of *Dixoniella* in EPS is well documented [20,36]; however, the potential of its biomass for the co-production of multiple products for a biorefinery approach has, to our knowledge, not yet been implemented. Our results show that *Dixoniella* represents a valuable bioresource for multiple products potentially useful as additives for lubricants: the EPS and cellular fraction, both rich in sugars and proteins, but different in their molecular composition of monosaccharides and amino acids. As a valuable side-product, we identified the lipid fraction, which harbors fatty acids and carotenoids. The data also show high variations between the ratios of the individual components and a clear dependency between the obtained product and the extraction technique used, which makes a standardized production process necessary. These findings are in accordance with several other studies, confirming that changes in culture conditions can influence the release and molecular composition of microalgae metabolites [89,90,91]. It is a common practice to induce the biosynthesis of target metabolites, such as pigments, EPS and lipids by changes in nutrient, salinity and light availability [22,69,74,92,93,94]. However, the production of multiple products would require further fine-tuning in order to meet the requirements of a multiple-product approach. Recent studies that focused on a biorefinery approach with different microalgae highlighted the interplay of nutrients, light and temperature to yield lipids, pigments and polysaccharides in a biorefinery concept [89,91,95]. The application of this process to *D. grisea* could be suitable for the harvesting of these promising metabolites to use as lubricants additives.

Although our hypothesis on finding multiple products as bio-additives in *D. grisea* is supported, further research is needed to generate a cost-effective process. Future work should therefore include follow-up cultivations to evaluate optimal culture condition for maximum multiple-product yield, which includes the examination of interactive effects of cultivation conditions on the synthesis of commercially relevant molecules. Furthermore, follow-up experiments are necessary to test the different culture fractions with regard to their physico-chemical properties in tribological processes.

## 4. Materials and Methods

### 4.1. Algae Growth Conditions

*D. grisea* strain UTEX 2320 was obtained from the culture collection of the University of Texas at Austin, purified to obtain axenic cultures, and was grown in BG-11 medium [96] supplemented with a commercial salt mixture (Instant Ocean, Aquarium Systems Technology AG, Grabs, Switzerland) to reach a final concentration of 8.25 psu and higher. BG-11 media was composed of 1.5 g NaNO_3_, 0.075 g MgSO_4_·7H_2_O, CaCl_2_·2H_2_O 0.036 g, citric acid 0.006 g, Na_2_CO_3_ 0.02 g, K_2_HPO_4_·3H_2_O 0.0525 g, ferric ammonium citrate 0.006 g, H_3_BO_3_ 0.00286 g, MnCl_2_·4H_2_O 0.00181g, ZnSO_4_·7H_2_O 0.000222 g, Na_2_MoO_4_·2H_2_O 0.000390 g, CuSO_4_ · 5H_2_O 0.000079 g, Co(NO_3_)_2_·6H_2_O 0.0000494 g and vitamin B_12_ (cyanocobalamin) 0.00002 g dissolved in 1 L of distilled water or salt mixture (see above).

Small 400 mL photobioreactors (FMT150, Photon Systems Instruments, Drásov, Czech Republic) were used for experiments with two different temperatures (20 °C and 25 °C) and using the same light and aeration regimes as described above. These photobioreactors were used at optimum temperature to study the growth with the addition of a carbon source (weekly addition of sodium bicarbonate to a final concentration of 3mM). All chemicals used were purchased from Carl Roth GmbH+Co.KG (Karlsruhe, Germany).

To provide enough working material for in-depth analysis of the different metabolites and the ecotoxicological screening, the cultures were grown in 1 L glass bottles (SCHOTT AG, DURAN^®^ GLS 80) under a 12:12 light:dark cycle (with fluorescent cool-white lamps, Ecolux F39W T5 865 ECO, GE lighting) at an irradiance of 150 µmol photons m^–2^ s^–1^, aerated continuously with sterile air at a flow rate of 100–150 mL min^–1^ and at room temperature (22 ± 5 °C). These cultures were used for the analysis of total monosaccharides, amino acids and fatty acids, as well as the ecotoxicological testing (see Section 4.4 and Section 4.5).

### 4.2. Sample Collection and Processing

#### 4.2.1. Cell Growth and Cell Counts

The optical density (OD) of the culture samples (2 mL) was determined at a wavelength of 750 nm using a UV/Vis spectrophotometer (Uvmini-1240, Shimadzu corp., Kyoto, Japan) in disposable semi-micro cuvettes (GBO, Frickenhausen, Germany). When the optical density was higher than 0.6, the samples were diluted with sterile media to reach a final OD_750nm_ measurement between 0.2–0.6. Dry weight and ash free dry weight were performed following Van Wychen and Laurens [97]. To overcome the inability to centrifuge the cultures due to their high viscosity, 5 mL of culture sample was first freeze dried and then burned in a muffle furnace (M110, Thermo Electron LED GmbH, Langenselbold, Germany). Cells counts were determined manually using a Neubauer chamber (Paul Marienfeld GmbH&Co, KG, Lauda-Königshofen, Germany) and diluted, if necessary.

#### 4.2.2. Analysis of Polysaccharides in Medium and Cells after Centrifugation (Medium-C, Cells-C)

*D. grisea* samples (1 mL sample with an OD_750nm_ between 0.2 and 0.5) were centrifuged at 15,000 rpm during 10 min at 4 °C (Mikro 200R, Andreas Hettich GmbH & Co. KG, Tuttlingen, Germany). Supernatant and cells were collected separately for further analysis and stored in the freezer (−20 °C). Quantification of sugars were measured separately, both in the supernatant (medium-C) and in the cells (cells-C). As the pigment content could interfere with the sugar analysis, cell samples were extracted with 1 mL of methanol 95% (Carl Roth GmbH+Co.KG, Karlsruhe, Germany) and placed overnight at 4 °C under subdued light. The day after, the extracted samples were centrifuged at 15,000 rpm during 10 min at 4 °C (Mikro 200R, Andreas Hettich GmbH & Co. KG, Tuttlingen, Germany) and the supernatant was used for pigment analysis following Zavrel et al. ([98], data not shown). The remaining pellet was used for sugar analysis after resuspension in 250 µL of distilled water. Sugar analysis was carried out following Zavrel et al. [99] with slight modifications in volumes: supernatant samples (250 µL) and cell samples (250 µL) were mixed with 250 µL of phenol 5% (Carl Roth GmbH+Co.KG, Karlsruhe, Germany) and were incubated for 15 min at room temperature. After that, 750 µL of sulfuric acid (Carl Roth GmbH+Co.KG, Karlsruhe, Germany) were added and incubated during 5 min. The samples were placed in a microplate (200 µL per sample) and measured in a microplate reader (Varioskan LUX, Thermo Fisher Scientific Inc., Walthamc, MA, USA) at 490 nm. Calibration was made using galactose as standard.

#### 4.2.3. Analysis of Lipids in Cells after Centrifugation (Cells-C)

Culture samples (1 mL) were collected and placed in the freezer (−20 °C) until further analysis. Lipid analysis was performed following Byreddy et al. [100] Briefly, 1 mL of homogenated sample was centrifuged at 15,000 rpm for 10 min at 4 °C (Mikro 200R, Andreas Hettich GmbH & Co. KG, Tuttlingen, Germany) dried and extracted with 1 mL of a mixture chloroform: methanol 2:1 *v*:*v*. The extracts were collected and dried at 100 °C for 10 min (Heatblock 2, VWR International GmbH, Darmstadt, Germany). Then, 2 mL of sulphuric acid was added and incubated at 100 °C for 10 min. After cooling at room temperature, 5 mL vanillin-phosphoric acid reagent (Carl Roth GmbH+Co.KG, Karlsruhe, Germany) was added for color developing and heated at 37 °C for 15 min (Thermomixer comfort, Eppendorf AG, Hamburg, Germany). After cooling at room temperature for 45 min in the dark, the samples were measured at 530 nm in the microplate reader described above. Calibration curve was made using commercial fish oil (Edeka, Omega-3 kapseln 1500 mg, Euro Vital Pharma GmbH, Hamburg, Germany) as standard.

#### 4.2.4. Analysis of Proteins in Medium and Cells after Centrifugation (Medium-C, Cells-C)

Protein analysis was performed using Folin–Ciocalteu reagent according to Lowry et al. [101] All chemicals were purchased from Carl Roth (Carl Roth GmbH+Co.KG, Karlsruhe, Germany). The method was as follows: Solution A (4 mg mL^−1^ NaOH and 20 mg mL^−1^ Na_2_CO_3_ in water) and solution B (10 mg mL^−1^ potassium sodium Tartrate and 5 mg mL^−1^ CuSO4 in water) were mixed at a 50:1 *v*:*v*. Samples were taken and separated as described for the sugar analysis. After centrifugation, supernatant was collected and 200 µL were used for the analysis. Pellet was resuspended in 200 µL MilliQ water. Then, samples were mixed with Lowry’s mixed solution as described above (mixture of solution A and B), vortexed and left for 15 min at room temperature. Finally, 100 μL 1.0 N Folin’s Phenol reagent (VWR International, Fontenay-sous-Bois, France) was added and left for 30 min for color development. A calibration curve was performed using bovine serum albumin (BSA, Carl Roth GmbH+Co.KG, Karlsruhe, Germany) as standard. The absorbance was measured in the microplate reader at 750 nm.

### 4.3. Viscosity

EPS solutions were prepared at a concentration of 0.5% by dilution of the lyophilized EPS in deionized water (25 mg in 5 mL) at room temperature and magnetic stirring overnight. To ensure complete dissolution and to disrupt potentially still-present aggregates, the solution was treated with ultrasound using a sonotrode (10 s, 50% amplitude, Bandelin Sonopuls HD 3100 with MS 72 tip, Bandelin electronic GmbH & Co. KG, Berlin, Germany). The viscosities were determined in triplicate at 25 °C with the rheometer Kinexus Pro (Malvern, Herrenberg, Germany) using a cone-plate geometry (1° angle and 50 mm diameter). The shear rate was varied between 1 to 1000 s^−1^ with 5 steps per decade.

### 4.4. Analysis of Total Monosaccharides, Amino Acids, and Fatty Acids

#### 4.4.1. Chemicals, Standards and Stocks

Used chemicals and materials were obtained as follows: hexane, methanol, acetonitril, 2-propanol all in gradient grade at Carl Roth Laborbedarf (Karlsruhe, Germany), pyridine water free (99.8%) and pyridine (≥99.9%) for syringe washing were purchased at Sigma-Aldrich (Steinheim, Germany). Helium and liquid nitrogen were delivered from Air Liquide (Bremen, Germany); solid chemicals and sugar standard substances were purchased at Sigma-Aldrich. *N*,*O*-bis(trimethylsilyl)trifluoroacetamide with 1% trimethylsilyl chloride (BSTFA + 1% TMSCl) and trifluoric acid were obtained at Supelco via Sigma-Aldrich.

Oleic Acid and a Supelco 37 Component FAME Mix GC Standard (Supelco 99.9%, p/n CRM47885). Trimethylsulfoniumhydroxid (TMSH) in methanol was obtained at Macherey-Nagel (Düren, Germany).

Norvaline and sarcosine were purchased at Sigma-Aldrich. Five amino acid standards (p/n 5061-3330–5061-3334), OPA reagent (conc.), (p/n 5061-3335) and FMOC reagent (conc.) (p/n 5061-3337) purchased from Agilent (Agilent Technologies, Santa Clara, CA, USA).

Stock solutions of all compounds and the working standards were stored in the deep freezer and were prepared daily by dissolving the standards in methanol or Chromasolv LC–MS Ultra water (Honeywell Riedel-de Haën, Fisher Scientific, Vienna, Austria) in amber vials.

#### 4.4.2. Extraction of *D. grisea* EPS and Cell Powder

Different batches of *D. grisea* cultured under laboratory conditions (see above) were used for the analysis of the different fractions (Table 1). The three fractions of complete cultures of the red alga *D. grisea* were EPS (EPS-P), the cell (cell-P), and the alcoholic media mixture itself (medium-P).

EPS were extracted by addition of 2-propanol 1:1 (*v*:*v*) to the culture batch with subsequent cooling (4 °C, dark) overnight and manual collection of the EPS after precipitation. This procedure was adapted from Khattar et al. [53] to reduce the total volumes in the extraction processes that would have to be processed later on for larger culture volumes > 1 L. Other reported extraction procedures use methanol or ethanol and higher volume ratios (2–3) [14,102]. The remaining media was either used as sample (=medium-P) or centrifuged at 10,000 rpm for 30 min at 4 °C (Sigma 3-18K, Sigma Laborzentrifugen GmbH, Osterode, Germany) to obtain cell samples (cells-P). A subset of medium-P samples was also dialyzed overnight in pure water (RC, pore size 3.5 kDa). All samples were lyophilized (Alpha 1-2 LD plus, Martin Christ Gefriertrocknungsanlagen GmbH, Osterode, Germany) prior to chemical analysis or ecotoxicological assessment.

#### 4.4.3. Quantification of Monosaccharides and Fatty Acids by GC-MSD as TMSE Derivatives

Gas chromatography was performed on a Hewlett Packard 6890 plus series GC-MSD (Agilent, Waldbronn, Germany). GC oven program was as follows: the oven was kept for 5 min at 100 °C followed by a ramp of 5 °C min^−1^ and a hold time at 300 °C. The capillary column was a 250 µm × 30 m FS Supreme-5ms with 0.25 µm film thickness purchased from CS (Chromatographie Service GmbH, Am Parir 27, 52,379 Langerwehe, Germany; Art. No.: 22545030). Column head pressure was kept constant at 1500 hPa. Helium (5.0) was used as carrier gas at a constant flow rate of 1.2 mL min^−1^. MS parameters: electron impact ionization (EI) at 70 eV, source temperature 230 °C, quadrupole 150 °C, full scan (50–400 amu).

Derivatization was performed with the dried samples as follows: 200 µL dried pyridine, 120 µL BSTFA + 1% TMSCl and 20 µL TFA were added into each vial. A total of 10 µL of oleic acid and erythritol in the concentration of 50 mg L^−1^ each in methanol were also added as external standards (Appendix A). All vials were crimped well, heated for 30 min at 70 °C and measured via GC-MSD. Response factors of the fatty acid and erythritol were used for quantification of all analytes. Response factors of the oleic acid, glucose and erythritol were used for the quantification of all analytes as trimethylsilyl derivates.

MS spectra were compared to the 8th NIST Edition MS Library (Agilent, Waldbronn, Germany) from 2008. Manual inspection of spectral matches and peak content further confirmed compound assignments and detected evidence of impurities or co-elution (Appendix A and Appendix A). Mass spectra comparisons with database entities were performed for each analyte. Only database match factors higher than 80% were used for comparison. For identification, the single ion mode (SIM) was used: for the hexoses *m*/*z* 204 and for pentoses *m*/*z* 217. The quantification was carried out with TIC signal areas.

#### 4.4.4. Quantification of Proteins by HPLC-DAD as Their OPA and FMOC Derivatives

The quantification of amino acids was conducted according to the methodology 5991-7922EN reported by Agilent (Agilent Technologies, Santa Clara, CA, USA) by an Agilent 1260 Infinity II Quaternary LC System: G7111B quaternary pump with WR G7115A Diode Array Detector (DAD), 10 mm flow cell, G7116A Multicolumn Thermostat and G7129A Vialsampler. The column was an AdvanceBio Amino Acid Analysis (AAA), C18 (4.6 × 100 mm, 2.7 µm, p/n 655950-802) with Guard column (p/n 820750-931) both purchased from Agilent. Due to the system pressure, chromatographic conditions reported in the methodology 5991-7922EN were carried out with some modifications: the flow speed was 0.8 mL min^−1^ with a column temperature set at 40 °C and injection volume of 1 µL. The mobile phases were 2.5 mM Na_2_HPO_4_ + 2.5 mM Na_2_B_4_O_7_, pH 8.2, as solvent A and acetonitrile:methanol:water (*v*:*v*:*v* 45:45:10) as solvent B. The elution gradient was as follows: 0–0.35 min 2% B, 0.35–13.40 min from 2% B to 57% B; 13.50–15.70 min 100% B; 15.80–18.00 min from 100% B to 2% B. For analysis of free amino acids in solution, a precolumn derivatization with injector programming of the HPLC’s autosampler using automated OPA and FMOC derivatization (OPA Reagent (conc.), p/n 5061-3335; FMOC Reagent (conc.), p/n 5061-3337) was used. The amino acid signals measured, correspond to the DAD signals at 338 nm (Ref = 390 nm) for the primary amino acids as OPA derivatives and at 262 nm (Ref = 324 nm) for the secondary amino acids as FMOC derivatives (Appendix A and Appendix A). To detect both OPA and FMOC-derivatized amino acids in one chromatogram, it was necessary to switch DAD wavelength. This switch took place between the last OPA derivative, lysine (19), and the first FMOC derivative hydroxyproline (20). For the extraction of amino acids, approximately 0.2 mg of EPS or pellet powder were mixed with 50 µL of HClO4 0.5 M, 100 µL internal standard solution, and 100 µL water. The mixture was sonicated at 42 kHz for 30 min at room temperature and then centrifuged at 4700 rpm and 4 °C for 10 min. The supernatant was filtered through a 0.2 µm polytetrafluoroethylene (PTFE) filter and immediately analyzed.

### 4.5. Eco-Toxicological Tests

#### 4.5.1. Chemicals, Standards and Stocks

All samples were weighed in after lyophilization, solved in the corresponding test medium, and stirred at least overnight before being used in a test. All samples containing algal cells or parts of them could contain also lipophilic substances (=cells-P and medium-P samples). Consequently, all such samples were tested twice: solved in the test media for testing of water-soluble substances as well as in a solvent-media mixture for lipophilic components. Dimethyl sulfoxide (DMSO, 99.9% purity; Sigma-Aldrich, Steinheim, Germany) was chosen as solvent due to its low toxicity [103,104,105,106]. A 1% *v*:*v* concentration was determined in pre-tests as the maximum concentration applicable for tests other than bacteria. The soil bacterium *A. globiformis* could be tested at 10% *v*:*v* due to its higher DMSO tolerance. To maximize the solution of lipophilic substances into DMSO, the algal powder was pre-dissolved at 10 g L^−1^ and stirred 3 d prior to use. This resulted in concentrations of 0.1 g L^−1^ of dried material for non-bacterial tests and 1 g L^−1^ of dried material for bacteria tests when samples were pre-solved in DMSO.

When medium was used for dissolution, algal samples were tested at 1 g L^−1^. In addition, daphnids were also tested at 0.6 g L^−1^ in case of EPS samples. Higher concentrations formed gel-like substrates that hindered the daphnids to maintain their breathing movements.

#### 4.5.2. Test Systems and Organisms

All chosen ecotoxicological tests were run in aquatic test media specifically suited for the chosen test organisms: Elendt M7 [107], artificial soil pore water E and C (Appendix A, respectively), and bacterial growth medium (Appendix A) for tests with the waterflea *Daphnia magna*, the potworm *Enchytraeus crypticus*, the springtail *Folsomia candida* and the soil bacterium *Arthrobacter globiformis*, respectively. Resources of the media salts are provided as an overview table in the supporting information (Appendix A). Pure medium served as the control for each sample.

#### 4.5.3. Ecotoxicological Screening

All tests were run according to standard protocols established in our laboratory. The test with the water flea *D. magna* was run according to the miniaturized protocol as described by Baumann et al. [108] based on the OECD guideline No. 202 [107]. The screening with the springtail *F. candida* was run using the water-only design described by McKee et al. [86] using a sea-salt medium as suggested by Houx et al. [109] and 8 replicates. The test with the soil bacterium *A. globiformis* was conducted according to Engelke et al. [110] The test with the potworm *E. crypticus* was established for this project by running a series of repeats with copper chloride as test substance using the aquatic design described in Roembke and Knacker [111]. These tests had EC50 values for Cu between 0.2 and 0.6 mg L^−1^ and a control mortality between zero and 10%.

For ecotoxicological screening, each algal sample was tested against a negative control, i.e., the medium without algal sample, and immobilization of test animals was compared. For the bacteria test, the enzymatic reduction of resazurin was used as endpoint and compared to the control. Samples were considered without negative effect when immobilization or enzymatic reduction did not exceed 10%.

For solvent-based samples, the comparison was run in relation to solvent control, containing the same amount of solvent as the sample’s test solution, but without the algal sample added. To ensure the validity of the tests, an additional medium only control was run, but not used for comparison.

The limited amount of sample did not allow to test more concentrations or to run repeats of each test. Consequently, no statistical analysis was performed for the screening.

For most batches, only some fractions (EPS, cells, medium) could be tested due to a lack of total material. For batch 9, the time used for EPS precipitation was too short to separate all EPS from the cells, so the cell and culture samples were excluded and only the obtained EPS was included in the analysis of results.

### 4.6. Statistical Analysis

All results with replicates were evaluated by descriptive and dispersion statistics, being the arithmetic mean and the corresponding standard deviations and standard errors. Statistical comparisons between treatments or batches were not performed in the analytical part, as the aim of this study was to provide an overview on the degree of variation of composition of biomass and exudated material between the different culture conditions. In the ecotoxicological part, differences between analyzed cultures were either strongly deviating and were considered to be outliers rather than observed effects, or, in case of the Collembola, variations were too high to detect any statistical differences.

For the establishment of the calibration of the analytical compounds via GC-MS and HPLC, linear correlation analysis was performed between the concentrations of the different sugars, fatty acids and proteins and their detected signals. To express the precision and repeatability of the used analytical methods the relative standard deviation (RSD) in % of all standard substances were specified. RSD in % is defined as the ratio of the standard deviation σ to the mean µ of all related area counts. All statistical analyses were carried out with the software Excel (MS Office, Washington, DC, USA).

## 5. Conclusions

In summary, we have shown that the molecular composition of *D. grisea* UTEX 2320 culture is suitable to obtain potential bio-additives for lubricants. We demonstrated that not only secreted polymers of red microalgae are valuable raw materials, but also the cells are important to consider for the production of bio-additives. We identified polysaccharides and proteins as the main components in most fractions and lipids as additional valuable composites. In particular, the functional groups of polysaccharides and proteins play an important role in the functionality of the molecules in tribological processes, which has already been proven in numerous studies and therefore supports our approach. We addressed the need for ecotoxicological testing, if algae-based molecules are to be approved as environmentally friendly materials. To this end, algal fractions derived from *D. grisea* were tested against aquatic and soil organisms and bacteria. The cellular fractions were not toxic, but the EPS of *D. grisea* requires further investigation, as it was harmful to enchytraeids.

Our results also indicate that culture conditions and analytical methodology impact the ratio and composition of macromolecules in *D. grisea*. Consequently, for commercial production of bio-additives, a standardized production and down-streaming process is necessary. The optimization of growth conditions in *D. grisea* would also enhance the production and availability of these metabolites. High viscosity cultures are a particular challenge for algae cultivation, but especially for biomass and media separation. Furthermore, from an industrial point of view, suitable extraction procedures and storage of the intermediates are the major bottlenecks that should be investigated in order to improve the yield and recovery of valuable algal products.

In conclusion, we identified multiple chemical intermediates in *D. grisea* cultures that might function as bio-additives in lubricants, and further investigations are necessary to (a) approve their techno-functional properties in the lubricating process, and (b) further formulate the intermediates into products.

## Figures and Tables

**Figure 1 plants-10-01836-f001:**
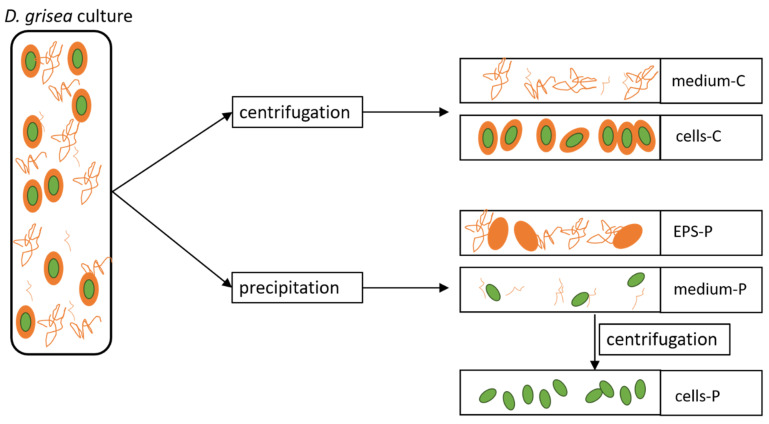
Overview on the different fractions analyzed for their composition. Fractions obtained by precipitation were also analyzed for their potential ecotoxicological effect.

**Figure 2 plants-10-01836-f002:**
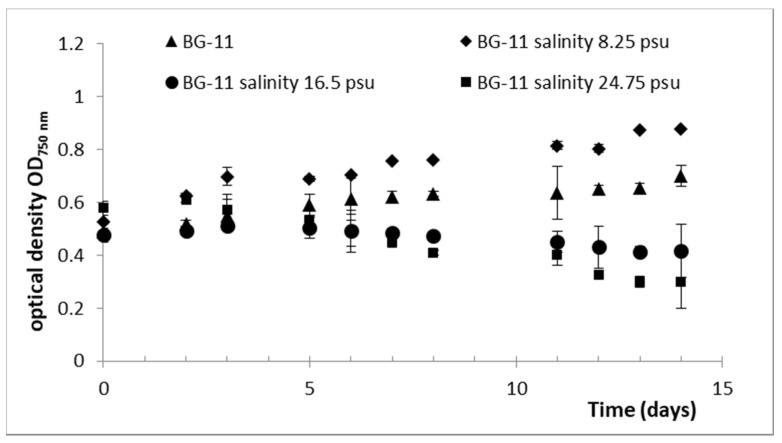
Growth of *D. grisea* UTEX 2320 in BG-11 freshwater medium with different salinities.

**Figure 3 plants-10-01836-f003:**
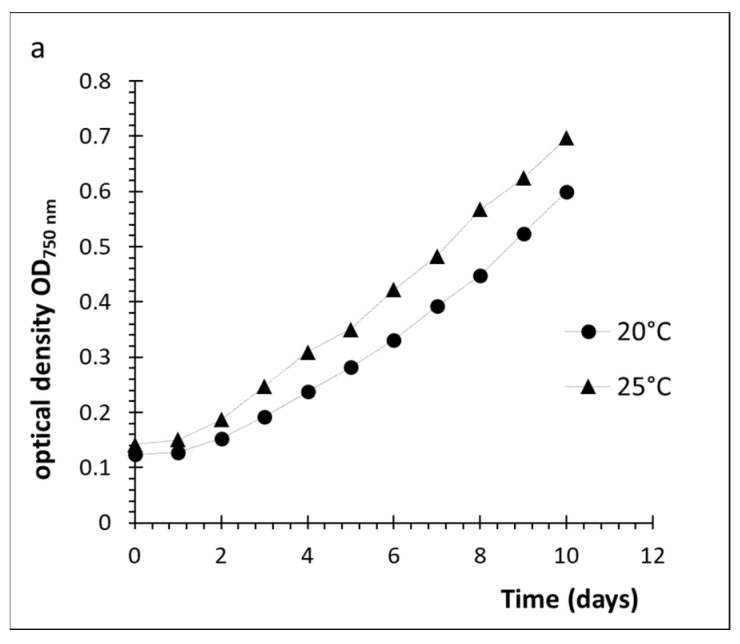
(**a**): Growth of *D. grisea* in BG-11 nutrients and a salinity of 8.25 psu under two different temperature conditions, 20 °C and 25 °C. (**b**): Polysaccharides (PS) and proteins released to the media at the end of the experiment. Values represent a mean of triplicates. Error bars represent standard deviations.

**Figure 4 plants-10-01836-f004:**
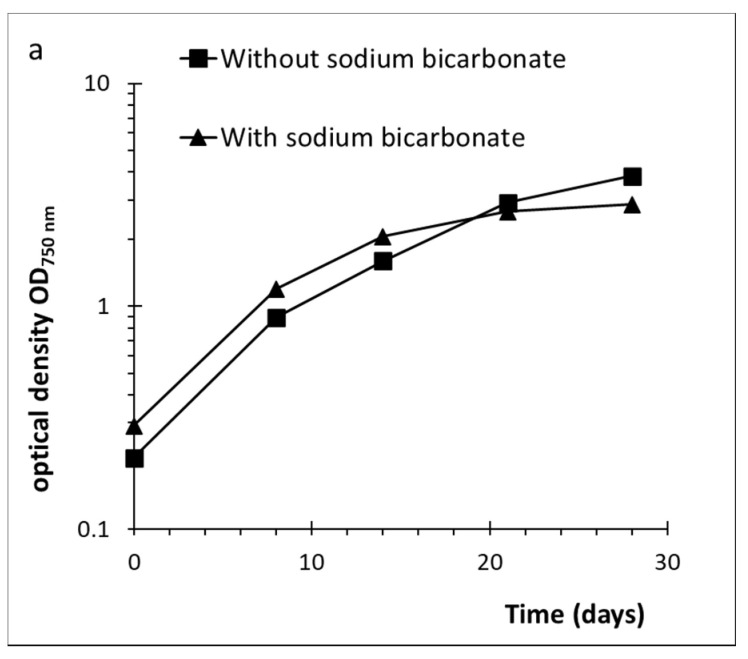
(**a**): Growth of *D. grisea* in BG-11 nutrients and a salinity of 8.25 psu at 25 °C with the addition of a carbon source (sodium bicarbonate final concentration 3 mM). (**b**): Cell counts per optical density 750 nm of *D. grisea*. (**c**): Percentage of polysaccharides (PS), proteins and lipids, related to ash free dry weight in the culture grown without additional sodium bicarbonate. For cellular and released PS and proteins proportions are given. (**d**): Percentage of polysaccharides (PS), proteins, and lipids related to ash free dry weight in the culture grown with sodium bicarbonate addition. For cellular and released PS as well as proteins, proportions are given.

**Figure 5 plants-10-01836-f005:**
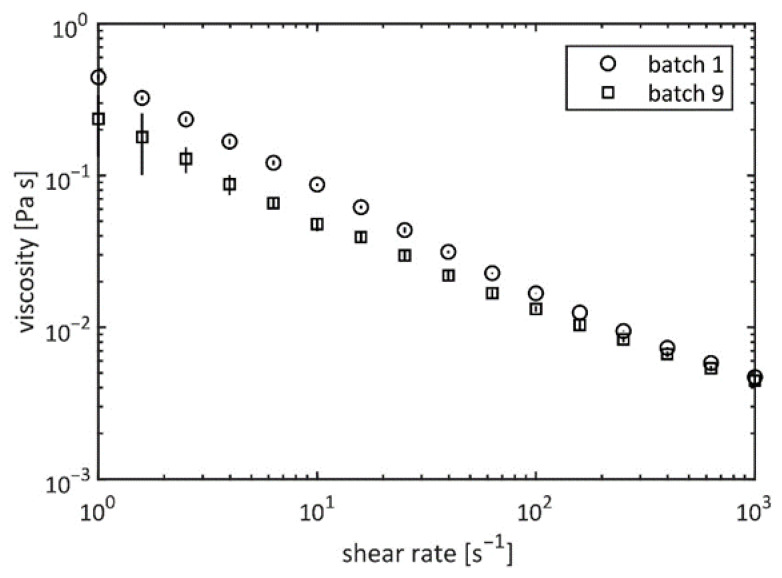
Viscosity measurements of the EPS in water at the concentration 0.5% (*w*/*w*) and 25 °C of the batches 1 and 9.

**Table 1 plants-10-01836-t001:** Overview on the cultured batches of *D. grisea* that were analyzed for monosaccharides, proteins, and fatty acids, and used in ecotoxicological tests, if remaining material sufficed for one replicate of each test. For each batch, the age of the culture at harvesting as well as any modification of the culture condition are presented.

Batch Number	Age of the Culture (Weeks)	Adaptation of Culture Condition
1	3	none
2	3	none
3	4.5	none
4	3	none
5	4	light intensity: 70 µmol photons m^−2^ s^−1^
6	4	additional 24 mM NaHCO_3_
7	4	none
8	4	none
9	1.5	none

**Table 2 plants-10-01836-t002:** Recovery rates for four internal standard substances according to the derivatization method and the complete sample preparation process of complete cultures of red alga *D. grisea*. Different microalgal batches were used for this determination. All recovery values are given as % of the corresponding substance. Three replicates and three repeats were used to determine the corresponding recovery rate.

		For Derivatization Only			For Total Work Up		
		EPS	MEDIA	CELLS	EPS	MEDIA	CELLS
Erythritol	TMSE Derivatization	90–96	73–76	86–91	0.5–2	93–97	0–2
Oleic Acid	TMSE Derivatization	93–98	72–76	74–78	3.5–5	14–20	63–71
Norvaline	OPA Derivatization	77–87	93–102	88–93	10	30.5	75
Sarcosine	FMOC Derivatization	89–93	94–98	94–104	19	79	39

TMSE: trimethylsilyl esters produced by the derivatization process; OPA: *o*-phthalaldehyde added by the derivatization process; FMOC: fluorenylmethoxycarbonyl produced by the derivatization process.

**Table 3 plants-10-01836-t003:** Overview of the GC-MSD analysis of the precipitated EPS (EPS-P) of *D. grisea* as quantification of total sugar content, of total fatty acid content, as well as the observed amount of glycerol. Additionally, the measured monosaccharides and the observed derivatives are given. All results are based on three replicates per batch.

Batch Number	Fatty Acids [mg g^−1^]	Glycerol [mg g^−1^]	Monosaccharides [mg g^−1^]	Single Monosaccharides	Monoasaccharide Derivatives
1	3.62 ± 0.04	4.52 ± 0.87	55.15 ± 6.09	Ara, Glc	Gal or Glc as oximes
3	24.63 ± 0.22	6.16 ± 0.95	40.33 ± 4.56	Ara, Rib, Xyl, Man, Glc	Gal or Glc alkylated or as alcohols and oximes
4	179.5 ± 1.79	Not detected	25.64 ± 2.74	Xyl	as alkylated sugar
5	93.26 ± 1.07	80.34 ± 8.57	25.82 ± 2.76	Man, Gal	alkylated or as alcohols
6	60.13 ± 0.69	184.7 ± 20.09	3.85 ± 0.42	Gal	Gal or Glc alkylated or as alcohols and oximes
7	96.51 ± 1.62	157.0 ± 16.68	124.42 ± 13.22	Rib, Gal	Gal or Glc alkylated or as alcohols and oximes
8	82.57 ± 1.34	94.8 ± 9.8	272.17 ± 28.21	Ara	Gal or Glc alkylated or as alcohols and oximes
9	70.55 ± 0.65	114.6 ± 13.0	68.01 ± 7.76	Xyl, Man, Gal	Gal or Glc alkylated or as alcohols and oximes

Ara: arabinose, Gal: galactose, Glc: glucose, Man: mannose, Rib: ribose, Xyl: xylose.

**Table 4 plants-10-01836-t004:** Quantification of total amino acid content, as well as the major amino acids identified in precipitated EPS (EPS-P) of *D. grisea*. All results are based on three replicates per batch.

Batch Number	Total Amino Acid Content [mg g^−1^]	Single Amino Acids
1	192.6 ± 0.46	Tyr, Lys, OH-Prol, Pro, Val
3	214.4 ± 0.52	Tyr, Lys, OH-Prol, Phe, Pro, Val, Ile
4	358.3 ± 0.75	Tyr, Cys, Lys, OH-Prol, Val, Ile
5	461.0 ± 0.68	Tyr, Lys, OH-Prol, Phe, Pro, Val, Ile
6	78.66 ± 0.19	Cys, Tyr, Lys, OH-Prol, Phe, Pro, Val
7	458.6 ± 1.54	Tyr, OH-Prol, Lys, Arg, Val, Pro, Ala, Phe, Ile
8	395.7 ± 0.68	Cys, Tyr, OH-Prol, Lys, Phe, Val, Pro
9	65.48 ± 0.13	Tyr, Lys, OH-Prol, Cys, Ile, Pro, Val

Ala: alanine, Arg: arginine, Cys: cysteine, Lys: lysine, Ile: iso-leucine, OH-Prol: hydroxyproline, Phe: phenylalanine, Pro: proline, Tyr: tyrosine, Val: valine.

**Table 5 plants-10-01836-t005:** Overview of the GC-MSD analysis of cells (cells-P) of *D. grisea* as quantification of total sugar content, of total fatty acid content, as well as the observed amount of glycerol. Additionally, the measured monosaccharides and the observed derivatives are given. In case of very low response during chemical analysis, a second set of samples was run. a/b: six samples were analyzed in total. To control the variability of every single analysis, three replicates were measured as one set on the same day (a) and a set of another three samples was measured as second set on a different day (b).

Batch Number	Fatty Acids [mg g^−1^]	Glycerol [mg g^−1^]	Monosaccharides [mg g^−1^]	Single Monosaccharides	Monoasaccharide Derivatives
1 ^a^	45.09 ± 0.31	313.7 ± 2.74	371.0 ± 3.24	Man, Gal, Glc, GlcUA	as alcohols
1 ^b^	184.5 ± 2.44	529.1 ± 7.01	199.0 ± 2.75	Man, Gal, Glc	as alcohols and oximes
2	21.81 ± 0.21	224.3 ± 2.53	448.6 ± 5.07	Ara, Gal, Glc, GlcUA	Gal or Glc alkylated or as alcohols and oximes
4	266.1 ± 2.55	304.7 ± 3.44	244.6 ± 2.76	Gal, Glc	Gal or Glc alkylated or as alcohols
5 ^a^	45.27 ± 0.31	267.5 ± 2.33	251.0 ± 2.19	Man, Gal, Glc	as alcohols and oximes
5 ^b^	80.56 ± 0.55	172.2 ± 1.51	255.0 ± 2.23	Gal	as alcohols and oximes
6 ^a^	93.43 ± 0.89	105.5 ± 1.19	50.17 ± 0.57	Ara, Man	as alcohols and oximes
6 ^b^	34.25 ± 0.33	270.6 ± 3.06	287.7 ± 3.25	Ara, Man, Gal	as alcohols and oximes
7	2.52 ± 0.02	20.71 ± 0.23	774.4 ± 8.7	Ara, Fuc, Gul	Gal or Glc alkylated or as alcohols and oximes
8	0.75 ± 0.01	106.5 ± 1.47	783.4 ± 10.8	Gal, Glc, GlcUA	Gal or Glc alkylated or as alcohols and oximes
9	8.88 ± 0.12	217.8 ± 3.01	544.4 ± 7.52	Ara, Gal, Glc, Fuc	Gal or Glc alkylated or as oximes

Ara: arabinose, Fuc: fucose, Gal: galactose, Glc: glucose, GlcUA: glucuronic acid, Gul: gulose, Man: mannose, Rib: ribose, Xyl: xylose.

**Table 6 plants-10-01836-t006:** Quantification of total amino acid content, as well as the major amino acids identified in cells (cells-P) of *D. grisea*. All results are based on three replicates per batch. In case of very low response during chemical analysis, a second set of samples was run. a/b: as more material was available for these batches, six samples were analyzed in total. To control the variability of every single analysis, three replicates were measured as one set on the same day (a) and a set of another three samples was measured as second set on a different day (b). The differences observed between the different analyses are within the range of variation of the different batches.

Batch Number	Total Amino Acid Content [mg g^−1^]	Single Amino Acids
1 ^a^	266.8 ± 0.58	Arg, Ala, Tyr, Asp, OH-Prol, Pro, Gln
1 ^b^	185.6 ± 0.44	Ala, Tyr, Asp, Glu, OH-Prol, Pro, Gln
2	364.2 ± 0.694	Lys, Arg, Met, Asp, OH-Prol, Phe, Ile, Glu
4	264.3 ± 0.59	Lys, Arg, Met, Asp, OH-Prol, Val, Ile, Phe, Leu
5 ^a^	368.3 ± 0.68	Lys, Arg, Met, Asp, OH-Prol, Phe, Leu
5 ^b^	497.7 ± 0.75	Arg, Ala, Tyr, OH-Prol, Lys, Val, Asp, Pro, Phe, Ile, Gly
6 ^a^	381.7 ± 0.70	Arg, Tyr, OH-Prol, Ala, Lys, Val, Pro, Asp, Phe, Ile, Gly
6 ^b^	457.7 ± 1.48	Arg, Tyr, OH-Prol, Ala, Lys, Val, Pro, Asp, Phe, Ile, Gly
7	561.9 ± 1.36	Tyr, Arg, OH-Prol, Lys, Ala, Asp, Val, Pro, Phe, Ile, Gly
8	273.9 ± 0.66	Lys, Arg, Met, OH-Prol, Asp, Phe
9	521.7 ± 3.24	Lys, Leu, OH-Prol, Asp, Met, Phe, Thr, Ile, Pro

Ala: alanine, Arg: arginine, Asp: aspartic acid, Cys: cysteine, Gln: glutamic acid, Glu: glutamine, Gly: glycine, Ile: iso-leucine, Leu: leucine, Lys: lysine, Met: methionine, OH-Prol: hydroxyproline, Phe: phenylalanine, Pro: proline, Tyr: tyrosine, Val: valine.

**Table 7 plants-10-01836-t007:** Results of the quantification of monosaccharides, amino acids, and fatty acid content in culture media with and without * dialysis of the alga *Dixoniella grisea* and the major components. All given results are based on three replicates per batch.

Batch Number	Fatty Acids [mg g^−1^]	Glycerol [mg g^−1^]	Monosaccharides [mg g^−1^]	Single Monosaccharides	Monosaccharide Derivatives	Amino Acids [mg g^−1^]	Single Amino Acids
5	49.1 ± 0.77	78.7 ± 1.68	588.2 ± 12.6	Ara, Man, Gal, Glc	as alcohols and oximes	461.5 ± 1.54	Lys, Arg, Met, Asp, OH-Prol, Phe, Leu
6	60.1 ± 1.01	119.1 ± 2.48	230.6 ± 4.81	Rib, Man, Gal, Glc	as alcohols and oximes	392.1 ± 0.86	Arg, Ala, Tyr, OH-Prol, Lys, Val, Asp, Pro, Phe, Ile, Gly
7	52.4 ± 0.82	89.6 ± 1.86	316.6 ± 6.77	Man, Gal	alkylated sugars only	368.7 ± 0.71	Tyr, Arg, OH-Prol, Lys, Ala, Asp, Val, Pro, Phe, Ile, Gly
5 *	366.6 ± 5.83	6.9 ± 0.45	14.62 ± 0.31	Man, Gal	alkylated sugars only	18.2 ± 0.03	Tyr, Lys, Arg
6 *	374.5 ± 5.95	9.1 ± 0.61	16.6 ± 0.35	Man, Gal	alkylated sugars only	11.8 ± 0.02	Tyr, Lys, Arg
7 *	435.0 ± 6.91	7.2 ± 0.53	9.2 ± 0.21	Man, Gal	alkylated sugars only	13.6 ± 0.02	Tyr, Lys, Arg

* indicate dialyzed samples in addition to the normal sample preparation. Ara: arabinose, Gal: galactose, Glc: glucose, Man: mannose, Rib: ribose. Ala: alanine, Arg: arginine, Asp: aspartic acid, Cys: cysteine, Gly: glycine, Ile: iso-leucine, Leu: leucine, Lys: lysine, Met: methionine, OH-Prol: hydroxyproline, Phe: phenylalanine, Pro: proline, Tyr: tyrosine, Val: valine.

**Table 8 plants-10-01836-t008:** Overview of the ecotoxicity of the screened lyophilized algal fractions obtained after precipitation sorted by batch used for extraction. Fractions containing EPS only were tested in the corresponding test medium only, fractions containing algal cells or parts of them were tested with medium and DMSO as solvent.

Batch No.	Fraction	Solvent	Concentration [g L^−1^]	Immobilized Daphnids ^1^	Immobilized Enchytraeids ^2^	Immobilized Collembola ^3^	Bacterial Enzyme Activity ^4^
1	EPS-P	Medium	1 ^a^	0.10	0.06 ± 0.13	0.30 ± 0.20	no effect
3				1.0	1.0 ± 0	0.50 ± 0.23	no effect
5				0	1.0 ± 0	0	no effect
7				0	1.0 ± 0	0	no effect
9				0	1.0 ± 0	0	n.r.
1	medium-P	Medium	1	0	0.31 ± 0.13	0	no effect
6				0	0	0	n.r.
7				0	0	0	n.r.
8				0	0	0	n.r.
1		DMSO	0.1 ^b^	0	0.06 ± 0.13	0.16 ± 0.17	no effect
6				0	0	0.03 ± 0.09	no effect
7				0	0	0.22 ± 0.36	no effect
8				0	0	0.06 ± 0.12	no effect
3	cells-P	Medium	1	0	1.0 ± 0	0	n.r.
5				0	0.06 ± 0.13	0	no effect
6				0	1.0 ± 0	0	no effect
3		DMSO	0.1 ^b^	0	0	0.19 ± 0.22	no effect
5				0	0	0.25 ± 0.38	no effect
6				0	0.06 ± 0.13	0.15 ± 0.24	no effect

^a^ for *D. magna* the concentration was 0.6 g L^−1^; ^b^ for *A. globiformis* the concentration was 1 g L^−1^; ^1^ values are presented as % of the total number of individuals (*n* = 10); ^2^ values are presented as mean and standard deviations (% individuals per replicate, *n* = 4); ^3^ values are presented as mean and standard deviations (% individuals per replicate, *n* = 8); ^4^ a qualitative scheme to indicate effects was used: positive, negative, or no effect; n.r. = runs without reliable results due to effects of the sample on the coloring agent.

## Data Availability

The data are contained in this article or in the supplementary material. Raw data are available for reasonable request from the authors.

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
