# Peer review of "Potential of the Red Alga Dixoniella grisea for the Production of Additives for Lubricants"

_plants, 2021, doi:10.3390/plants10091836_

Round 1

Reviewer 1 Report

A marine red microalga Dixoniella grisea was used for the application of the exocellular polymers as environment-friendly bio-additives. Under different salinities and temperatures, the growth of algal cells and the content of different fractions were examined. 

The methods ate solid, but the statistical analysis is not described. 

Author Response

Dear reviewer,

many thanks for your constructive and supporting feedback to our manuscript. We altered the corresponding parts according to your suggestions. An overview of the changes made is provided below.

You asked for a description of the statistical methods used. We improved the corresponding section (see below). In addition, we like to point out that the submitted manuscript focuses on a descriptive analysis rather than an experimental comparison of effects. As a consequence, statistical analysis such as ANOVA or regression could not be run for the dataset collected in this study. Instead, mean values and standard errors were used to provide a first overview about the degree of variation caused by the different analytical methods and culture conditions. The authors are aware that this can only represent a first step towards a hypothesis-driven experimental design. However, we consider this descriptive study necessary to illustrate the potential uses of red algal EPS as well as to provide a basis for designing cause-effect-studies.

Overview on changes made on the manuscript:

Comment 1, reviewer 1:

The methods ate solid, but the statistical analysis is not described.

>    Line 864 – 878:

All results with replicates were evaluated by descriptive and dispersion statistics, being the arithmetic mean and the corresponding standard deviations and standard errors. Statistical comparisons between treatments or batches were not done in the analytical part, as the aim of this study was to provide an overview on the degree of variation of composition of biomass and exudated material between the different culture conditions. In the ecotoxicological part, differences between analyzed cultures were either strongly deviating and were considered to be outliers rather than observed effects, or, in case of the Collembola, variations were too high to detect any statistical differences.

For the establishment of the calibration of the analytical compounds via GC-MS and HPLC, linear correlation analysis was performed between the concentrations of the different sugars, fatty acids, and proteins and their detected signals. To express the precision and repeatability of the used analytical methods the relative standard deviation [RSD] in % of all standard substances were specified. RSD in % is defined as the ratio of the standard deviation σ to the mean µ of all related area counts. All statistical analyses were done with the software Excel (MS Office, USA).

Reviewer 2 Report

The study is about the potential of the red alga Dixoniella grisea for the production of lubricants reported by Olea and Siol, and coworkers. The revision has a significant improve compared with the original version. However, before it will be suggested for publication in Plants, the following issue should be well addressed.

  1. Line 75, several recent studies (doi.org/10.1016/j.actbio.2018.11.015; doi.org/10.3390/lubricants9030027; doi.org/10.1021/acs.langmuir.9b02935) should be included to support such claim.
  2. Line 75-77, in the thin film lubrication mode, the viscosity of the lubricants is very important, but it is not necessary for boundary lubricants. The authors should make this point clearer.
  3. The authors should add some discussion about why such extraction method (2-propanol 1:1 (v:v)) was used.

Author Response

Dear reviewer,

many thanks for your constructive and supporting feedback to our manuscript. We altered the corresponding parts according to your suggestions. An overview of the changes made is provided below.

Thank you for the more recent sources. By adding more details in the Introduction to the reported lubrication mechanisms we hope to make the role of viscosity of the lubricating film for the lubrication more clear. Additionally, we added an explanation for the use of 2-propanol as precipitation agent in the volume ratio 1:1 (v:v) in the Materials and Methods section.

Comment 1 and 2, reviewer 2:

Line 75, several recent studies (doi.org/10.1016/j.actbio.2018.11.015; doi.org/10.3390/lubricants9030027; doi.org/10.1021/acs.langmuir.9b02935) should be included to support such claim.

> We greatly acknowledge reviewer suggestion about recent works, but, regarding “doi.org/10.1021/acs.langmuir.9b02935” (Scarratt et al. 2020) we think that it is a little bit out of our paper focus.

Line 75-77, in the thin film lubrication mode, the viscosity of the lubricants is very important, but it is not necessary for boundary lubricants. The authors should make this point clearer.

>    We changed the corresponding paragraph to:

Lines 72 – 95:

A biotechnologically important source of algae-based PS are species of cyanobacteria, diatoms and green and red microalgae. Exopolysaccharides of the cyanobacterium Cyanothece epiphytica showed excellent potential as biolubricant [14]. This was related to the similarity of the measured visco-elastic properties of the EPS to conventional grease, showing a high storage modulus compared to the loss modulus (G’ >> G’’). These properties are supposed to stabilize the lubricant film thickness when high pressures occur, e.g., in rolling bearings of a high load [14]. High viscosities are considered favorable for lubrication as viscosity controls the lubrication film thickness (Paul et al. 2021). Arad et al. (2006) state that at high pressures, high loads and low sliding velocities the main friction mechanism is boundary lubrication. They also stressed that adhesion of red microalgae polysaccharides, that was related to also present glycoproteins, was an important advantageous influence on the lubrication compared to the properties of hyaluronic acid alone. Interestingly, strong lubricating boundary layers were reported by Lin et al. (2019) when using hyaluronic acid together with phosphatidyl choline lipids for tendons. This strong effect was related to the also present glycoprotein lubricin. The study [14] also corroborated the versatility of exopolysaccharides, showing their great potential as emulsifier, flocculant and disperser. Gasljevic and coauthors [15] evaluated the polysaccharides of several marine microalgae as suitable drag-reducing additives for naval applications. They found that the red microalgae species Porphyridium cruentum and Rhodella maculata, and the green microalgae species Schizochlamydella capsulata and Chlorella stigmatophora exhibited the best drag-reducing ability among the strains tested. They also included cellular polysaccharides into their study, which revealed similar properties than the extracellular polymers (EPS), and when applied together, increased drag-reducing ability by fourfold [15]. The potential of PS from red microalgae in tribological processes was superior to the conventional hyaluronic acid as a lubricant in terms of friction reduction, adsorption and stability [24,25]. Notably, only low polymer concentrations were necessary to result in high viscosity [26].

Comment 3, reviewer 2:

The authors should add some discussion about why such extraction method (2-propanol 1:1 (v:v)) was used.

>    We added a reference to the last paragraph of the Introduction and modified in the Materials and Methods section the paragraph related to the extraction:

Lines 150-152:

Second, precipitation of EPS was induced by addition of 2-propanol following Khattar et al. [53] using a ratio of 1:1 (v:v) (see details Materials and Methods section).

Lines 754-763:

EPS were extracted by addition of 2-propanol 1:1 (v:v) to the culture batch with subsequent cooling (4 °C, dark) overnight and manual collection of the EPS after precipitation. This procedure was adapted from Khattar et al. [53] to reduce the total volumes in the extraction processes that would have to be processed later on for larger culture volumes > 1 L. Other reported extraction procedures use methanol or ethanol and higher volume ratios (2–3) [14, 103]. The remaining media was either used as sample (= medium-P) or centrifuged at 10,000 rpm for 30 min at 4 °C (Sigma 3-18K, Sigma Laborzentrifugen GmbH, Osterode, Germany) to obtain cell samples (cells-P). A subset of medium-P samples was also dialyzed overnight in pure water (RC, pore size 3.5 kDa). All samples were lyophilized (Alpha 1-2 LD plus, Martin Christ Gefriertrocknungsanlagen GmbH, Osterode, Germany) prior to chemical analysis or ecotoxicological assessment.

This manuscript is a resubmission of an earlier submission. The following is a list of the peer review reports and author responses from that submission.

Round 1

Reviewer 1 Report

Gavalás-Olea and coworkers investigated the red alga Dixoniella grisea for production of additives as lubricants. The results are not well organized and well presented. The following issues should be address before it could be considered for publication in Plants.

  1. The quality of the figures (figure 1-3) should be improved. I suggest the authors could learn it from the following paper (Plants 2021, 10(1), 158; https://doi.org/10.3390/plants10010158), which was published in Plants.
  2. The introduction provides a brief background, but the tribology section is weak.
  3. What’s the exact molecular composition of extra- and intracellular from the red microalga Dixoniella grisea? I can’t see it clearly from the abstract or conclusion.
  4. There is no actually tribological study in this ms. I don’t understand why the authors could claim these extracts have potential application as lubricants.
  5. The resources of all the chemicals used should be listed.
  6. Format issues: MgSO4*7H2O should be changed to MgSO4‧7H2O (please correct all).

Reviewer 2 Report

The manuscript entitled "Potential of the red alga Dixoniella grisea for the production of additives for lubricants" is relatively well written and has an interesting goal. However, as it stands, I cannot recommend its publication.

Major issues:

Figs. 1 and 2 – the authors should have monitored growth using dry weight instead of OD, unless they presented a calibration curve between OD and dry weight; OD can be affected by many factors and is a poor measurement of actual growth in a microalgal culture, in particular if the strain is known to secrete EPS and growth is not done under axenic conditions.

Figs 2a, 3b.  – I find the results of the protein, polysaccharide and lipids as shown in this figure highly questionable. This would mean that the cells would have a composition as something like: 58 % of PS, 41% protein and <1% lipids (not taking into account ashes, etc.). This cannot be true at all. The minimum lipid contents I've seen in microalgae, including Rhodophyta, is ≈7%. The authors are highly advised to recheck their biochemical data, as there is a major problem with the way they quantified the lipid contents.

Page 6 – the authors very often give values in %. However, they never state of what. Cases in point: lines 290, 293, 294, etc. When you give these values you need to state % of something.  

Tables 3 and 4 – I do not understand the differences found in the first three replicates and the last three replicates. The authors need to help the reader to understand those differences and if they are acceptable or not. Moreover, if they are means of triplicates the standard error should be given as well.

Minor issues:

Line 41 – where it reads "potential, for example proven", it should read "potential as, for example, proven"

Line 44 – To motivate is a transitive verb and therefore needs an object. Who are you motivating to do what?

Line 48 - where it reads "industry", it should read "industries". Moreover, do not mix e.g. with for, as this Latin locution does not require such a preposition, but does require to be between parentheses.

Line 50 – put "oil or aqueous" before the noun that is qualifying, i.e., "liquid".

Lines 50-52 – please rephrase by avoiding the use of lubricants more than once. Moreover, avoid starting a sentence with a conjunction, i.e., "and". You can use it in colloquial English, but please avoid that in formal English. Instead use expressions like "Moreover", "In addition", etc. Please check for this style error throughout the manuscript.

Line 66 – "living benthic" is an incorrect expression. Please revise. The proper usage would be something like "living in the benthic zone."

Line 87 – please avoid starting a sentence with "due to". In addition, "due to" has a meaning similar to that of "caused by". Use "because of" instead. This error is common, though, in the literature.

Lines 101-102 – "render different chemical properties" – this is vague; please state exactly which chemical properties are different.

Lines 103-106 – please break the sentence into two; avoid connecting two sentences with "however"; start a new sentence with it, instead.

Line 123 – remove comma after "both".

Line 125 – please use "sector" in its plural form.

Lines 125-126 – I do not understand what the authors mean with "highest experience". Please rephrase.

Line 238 – Although "AA" was introduced in line 10, please avoid its use, as that decreases de readability of the manuscript, because its use is only repeated in another section. The same recommendation is given for "FA" and other abbreviations which are not immediately used after a few sentences. For example, in line 278, I am at a loss what AS meant.